# Numerical Simulation Study on the Law of Attenuation of Hydrate Particles in a Gas Transmission Pipeline

**Rao Yongchao** [1,2]**, Sun Yi** [1,2]**, Wang Shuli** [1,2,*] **and Jia Ru** [1,2]

[1]   Jiangsu key laboratory of oil-gas storage and transportation technology, Changzhou, Jiangsu 213164, China; ryc@cczu.edu.cn (R.Y.); 17000383@smail.cczu.edu.cn (S.Y.); 17000419@smail.cczu.edu.cn (J.R.)

[2]   Changzhou University, School of petroleum engineering, Changzhou, Jiangsu 213164, China

[*]   Correspondence: wsl@cczu.edu.cn; Tel.: +86-13813698610

**Abstract:** Based on the swirl flow of gas hydrate pipeline safety flow technology, the numerical simulation method is used to study the attenuation law of hydrate particles, which is of great significance for expanding the boundary of safe flow. The results show that the size of the initial swirl number is mainly related to the twist rate and has nothing to do with the Reynolds number; the smaller the twist rate, the greater the Reynolds number, the greater the number of swirling flow in the same position in the pipeline. The concentration has almost no effect on the change of the swirl number; for the non-dimensional swirl number, and the numerical simulation is roughly the same as the results of the paper, the attenuation coefficient beta and ln (Re) has a linear relationship. The no twist tape is six to eight times larger than the volume fraction of the twisted belt, and the smaller the twist tape twist, the smaller the particle deposition is, the higher the initial concentration of the particles in the pipe, and the larger the volume fraction of the hydrate particles deposited by the tube wall.

**Keywords:** hydrate particles; safe transportation; swirl flow attenuation; swirl number; numerical simulation

---

## 1. Introduction

In the 1930s, American scientist, Hammerschmidit [1], first discovered hydrates in pipelines, which confirmed that natural gas hydrates are the cause of blockage of pipelines and are getting more and more attention from the outside. Natural gas hydrates control technology mainly includes heating, pressurization, and adding chemical reagents [2]. However, there are disadvantages, such as high cost, high risk, and a polluted environment. Therefore, the gas pipeline hydrates' blockage prevents and controls changes of the traditional complete inhibition to the risk management, which ensures the safe flow in the gas pipeline [3–7].

Swirl flows refer to the rotational motion formed by the superposition of tangential velocity while the fluid moves in the axial direction [8]. Swirling is part of the recognized forms of flow in industrial equipment and is valued for its wide range of applications and complex scientific foundations. The attenuation characteristics of swirl flow have gradually attracted research worker's attention to engineering practice, such as the isolator, combustion chamber, and heat exchanger. Wang and Yongying [9] deduced the swirl flow attenuation formula for the straight blade and the wrong blade of the combustion chamber and found that the two formulas have similarity. Wu and Linpeng [10] used a multi-blade type spinner to form a swirl flow conveniently and efficiently, and point out that the swirl component in the swirl flow is easily attenuated and needs special facilities to maintain it. Kreith [11] showed that for the swirling of fixed tubes, the attenuation of swirling intensity is mainly affected by the Reynolds number, and is not affected by the intensity of the inlet swirl. Kitoh [12]

observed the influence of the Reynolds number on different swirl intensity ranges, and obtained the correlation between the swirl intensity range and Reynolds number. Steenbergen [13] found through experimental research that although the difference in initial conditions still exists in a large part of the decay process, the decay rate is not substantially different. Zhang and Lili [14] experimentally studied the swirl flow attenuation characteristics of a flat-axis tube produced by a local spinner. The experiment concluded that the tangential velocity and the swirling intensity decrease with an increasing distance.

The swirl flow hydrates control technology was first proposed by the hydrate research group of Changzhou University. The swirl flow characteristics include velocity distribution, pressure drop distribution, and swirl to flow attenuation characteristics. At present, there is still little research on the swirl flow attenuation law. By studying the swirl flow attenuation characteristics in natural gas pipelines, the effects of swirl gas flow on hydrate particles are revealed to provide a guarantee of the safe flow of hydrates.

## 2. Model Establishment

### 2.1. Physical Model

A horizontal pipeline with a diameter, $D = 0.024$ m, and length, $L = 2.5$ m, as shown in Figure 1, was established. The pipe is divided into two pipe segments, before and after. The front pipe segment contains a twisted band structure of a length $L_1 = 0.5$ m. The twisted belt is set at the entrance into the pipeline, and three different twist ratios of 6.2, 7.4, and 8.8 were set, respectively. The geometric model of the twisted belt is shown in Figure 2. For the entire geometric model meshing, the front pipe segment uses an unstructured mesh to encrypt the mesh at the pipe wall. The calculation grid is illustrated in Figure 3. The rear tube segment adopts a structured grid, which can reduce the error of numerical simulation and improve the computational efficiency of the grid. After the grid independence test, the number of grid cells was approximately one million. The origin of the three-dimensional coordinate system of the calculation model is at the center of the entrance interface, and the fluid medium is natural gas. The Z-axis is the flow direction. The Y-axis is the gravity direction. The Euler model was utilized to simulate the gas-solid two-phase three-dimensional swirl flow of the gas pipeline, using energy equations, pressure bases, and implicit solvers.

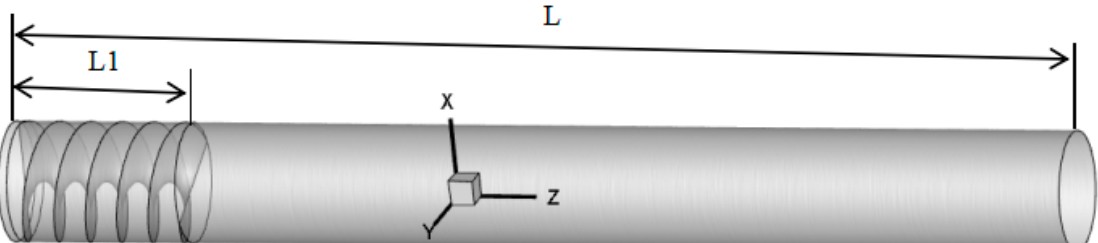

**Figure 1.** Calculation model.

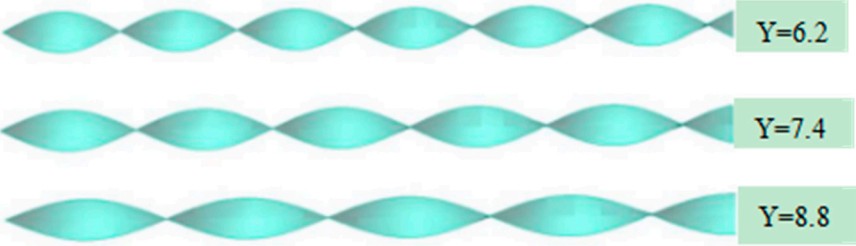

**Figure 2.** Twist tape model.

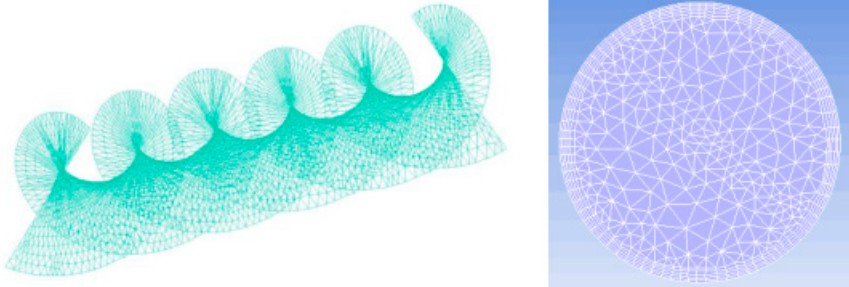

**Figure 3.** Grid model.

*2.2. Mathematical Model*

Using ANSYS calculation software, FLUENT 16.0(the most powerful computational fluid dynamics (CFD) software available.), the Euler model was used to simulate the gas-solid two-phase three-dimensional swirl flow of the gas pipeline, and the energy equation was selected. The pressure base and the implicit solver were accustomed. The continuous phase flow field calculation uses the finite element volume method, the pressure-velocity coupling method uses the SIMPLEC (Semi-Implicit Method for Pressure-Linked Equations Consistent) algorithm, and the turbulence model uses the RNG (Reynolds-Normalization Group) *k-ε* type. The pressure needs a multi-dimensional linear reconstruction method to reconstruct the second-order format of surface pressure, and the particle motion model uses the DPM (Discrete Phase Model) model. The selection of the downward relaxation factor of the iterative process takes values of the following ranges: $\varepsilon_p = 0.3{\sim}0.7$, $\varepsilon_m = 0.5{\sim}0.7$, and $\varepsilon_k = \varepsilon_\varepsilon = 0.4{\sim}0.6$. The convergence condition as the absolute value of the residual $<1 \times 10^{-6}$ was defined. Next, the governing equations were used:

Continuity equation:

$$\frac{\partial \rho}{\partial t} + \frac{\partial}{\partial x_i}(\rho u_i) = 0, \tag{1}$$

Momentum equation:

$$\frac{\partial}{\partial t}(\rho u_i) + \frac{\partial}{\partial x_j}(\rho u_i u_j) + \frac{\partial p}{\partial x_i} - \frac{\partial \tau_{ij}}{\partial x_j} - \frac{\partial \tau_{ij}^{-1}}{\partial x_j} = 0, \tag{2}$$

where, $\rho$, $u$, and $P$ are the gas density, velocity, and static pressure; $\tau_{ij}$ is a viscous stress tensor; and $t$ is time.

Volume fraction equation:

$$\frac{1}{\rho_q}\left[\frac{\partial}{\partial t}(a_q \rho_q) + \nabla \cdot \left(a_q \rho_q \vec{v_q}\right) = S_{aq} + \sum_{\rho=1}^{n}\left(\vec{m}_{pq} - \vec{m}_{qp}\right)\right], \tag{3}$$

where, $a_q$ is the q-phase volume fraction, which is the source term. Additionally, $\vec{m}_{pq}$ it is the mass transfer mass of the *p*-phase to the *q*-phase, and $\vec{m}_{qp}$ is the mass transfer of the *q*-phase to the *p*-phase.

Energy equation:

$$\frac{\partial(\rho T)}{\partial t} + \frac{\partial(\rho u T)}{\partial x} + \frac{\partial(\rho v T)}{\partial y} + \frac{\partial(\rho w T)}{\partial z} = \frac{\partial}{\partial x}\left(\frac{\lambda}{c_p}\frac{\partial T}{\partial x}\right) + \frac{\partial}{\partial y}\left(\frac{\lambda}{c_p}\frac{\partial T}{\partial y}\right) + \frac{\partial}{\partial z}\left(\frac{\lambda}{c_p}\frac{\partial T}{\partial z}\right), \tag{4}$$

where, $\rho$, $c_p$, and $T$, are, respectively, the gas density, constant pressure, specific heat capacity, temperature, and thermal conductivity; $u$, $v$, and $w$ are the speed; and $t$ is the time. The DPM model obtains the particle motion equation by calculating the force applied by the particle, and determines

the motion trajectory of the motion equation of the particle under Lagrange coordinates. The equation of motion of the solid particle trajectory in the z-direction is [15]:

$$\frac{du_p}{d_t} = F_D(u - u_p) + \frac{g_z(\rho_p - \rho)}{\rho_p} + F_z,$$ (5)

$F_D(u - u_p)$ is the unit mass drag of the particles, (N/m$^2$); $\frac{g_y(\rho_p - \rho)}{\rho_p}$ is the difference between gravity and buoyancy of the particles; $g_z$ is the component of gravity acceleration in the z-direction; $F_z$ represents the other forces of the particles; u is the fluid phase velocity, (m/s); $u_p$ is the particle speed, (m/s); $\rho_p$ is the particle density, (kg/m$^3$). The other forces, $F_z$, acting on the particles mainly include the acceleration of the fluid around the particles, the additional mass force, the Brownian force, the Staffman rising buoyancy, etc. The additional mass force is mainly used when the fluid phase density is greater than the particle density. The hydrate particle density in the pipe is greater than the airflow density, so there is no need to consider the additional mass force. $F_D$ is defined as:

$$F_D = \frac{18\mu}{\rho_p d_p{}^2} \frac{C_D \text{Re}}{24},$$ (6)

Re is defined relative to the Reynolds number:

$$\text{Re} = \frac{\rho d_p |u_p - u|}{\mu},$$ (7)

The $d_p$ is the particle diameter, (m); the drag coefficient, $C_D$, was set to constant, and the value was 0.44. The expression of Brownian force is:

$$F_B = \zeta \sqrt{\frac{\pi S_0}{\Delta t}},$$ (8)

The $\zeta$ is a Gaussian random function with a mean of zero and a variance of 1. $\Delta t$ is the time step of the particle. $S_0$ is the spectral intensity function.

### 2.3. Boundary Conditions

The pipe diameter was $D$ = 0.024 m, and the pipe length was $L$ = 2.5 m. The gas phase was set to natural gas. The natural gas density was 0.717 kg/m$^3$, and the natural gas kinematic viscosity as $11.03 \times 10^{-6}$ m$^2$/s. The inlet boundary adopts the speed inlet condition, and the four Reynolds numbers Re were selected as 5000, 10,000, 15,000, and 20,000, respectively. The outlet boundary used outflow conditions. The inlet temperature, Tin, was 280 K, and the wall temperature was $T_w$ = 277 K. It was considered that the physical properties of the hydrate particles were constant, the particle density was 915 kg/m$^3$, and the diameter was 0.1 mm. The initial concentration of hydrate was set to 1%, 2%, 4%, 6%, and 8% in DPM. Considering the gravitational boundary conditions between gravity and the gas-solid two phases, the problem was reduced to a three-dimensional, atmospheric gas-solid two-phase flow.

### 2.4. Grid Independence Verification

To meet the calculation accuracy requirements and to ensure the calculation efficiency, three grid sizes of 8 mm, 4 mm, and 2 mm were selected for grid independence verification. The simulated verification conditions were: Pipe diameter was $D$ = 24 mm; gas Reynolds number, Re, was 20000; the twist band twist rate was 6.2; and the initial particle concentration was 2%. As shown in Figure 4, the velocity distribution on the vertical line of the cross-section at the pipe, $Z$ = 5 $D$, was selected for comparison. Under the three grid sizes selected, the obtained velocity distributions had similar trends. However, compared to 2 mm (150,948 cells) and 4 mm (116,570 cells), the mesh size was 8 mm (654,120 cells), and the calculation accuracy was not enough due to the small mesh at the near wall. The mesh size of 2 mm (150,948 cells)

and 4 mm (116,570 cells) were the same, and the accuracy of the near wall was not improved. Therefore, the grid size of 4 mm (116,570 cells) was selected as the calculation grid.

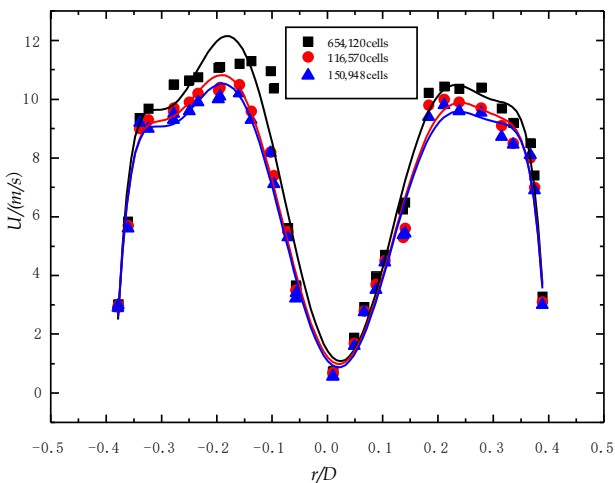

**Figure 4.** Velocity distribution in the different sections.

### 2.5. Model Verification

The numerical simulation results of the hydrate pipeline swirl flow and experimental data were compared and verified. Figure 5 shows the comparison of the verification results. The pressure drop of the fluid changed in the Reynolds number (*Re*) in the gas-solid two-phase flow. The pipe length was 2.5 m, the pipe diameter was 24 mm, the particle size was 0.1 mm, the twist band twist ratio was 6.2, and the initial particle concentration was 2%. The Reynolds number Re was calculated by the formula, $Re = vd\rho/\mu$, where $d$ is the diameter of the pipe, $v$ is the average velocity of the hydrate particles, and $\rho$ and $\mu$ are the density and dynamic viscosity of the gas. As shown in Figure 5, it can be seen that the error between the simulation result and the experimental result was small. The verification results show that the simulation calculation was reliable.

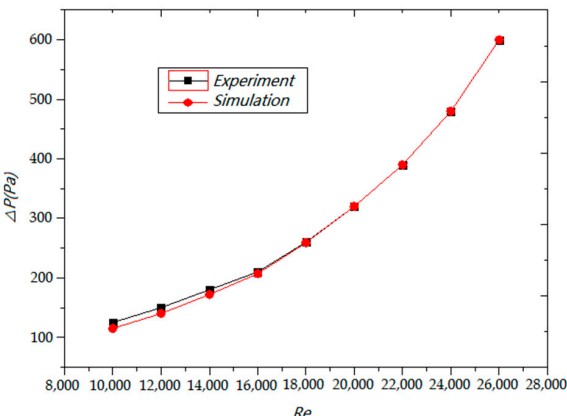

**Figure 5.** Pressure drop with different *Re*.

## 3. Results and Discussion

### 3.1. Swirl Number

#### 3.1.1. Definition

The swirl number, *S*, is an important physical quantity characterizing the flow characteristics of the swirl flow, which can characterize the degree of swirling flow in the swirl flow, and is usually

defined by the magnitude of the circumferential speed or the angular velocity of rotation. In the hydrate particle transport of the gas pipeline, the influence of the swirl flow on the hydrate particles is mainly determined by the magnitude of the swirling force compared with the non-swirl flow. Therefore, the swirl number was studied to obtain the swirl flow of the hydrate particles. The law of attenuation is thus of great significance.

The following is a derivation of the swirl number in a gas-solid two-phase swirl flow. The swirl number in a single-phase swirl flow is defined as the number of dimensionless swirl, and the formula is:

$$S = \frac{\int_0^R u_\theta u_a r^2 dr}{R \int_0^R u_a{}^2 r dr}, \tag{9}$$

where, $u_a$ is the axial velocity, $u_\theta$ is the tangential velocity, and $R$ is the pipe diameter. It was considered that the turbulent disturbance (pulsation) was negligible. Therefore, the velocity was considered to be the average speed.

### 3.1.2. The Two-Phase Flow of Gas-Solid Swirl Number Calculation Formula

Based on the swirl flow model of the gas-solid two-phase pipeline, as shown in Figure 6, the number of single-phase swirls defined by the simultaneous equation (9), and the swirl number, $S$, of the swirl flow of the gas-solid two-phase pipeline were defined as follows [16]:

$$\begin{aligned}
S &= \frac{\int_0^{R_g} \rho_g u_{ag} u_{\theta g} r^2 dr + \int_{R_g}^R \rho_s u_{as} u_{\theta s} r^2 dr}{R(\int_0^{R_g} \rho_g u_{ag}{}^2 r dr + \int_{R_g}^R \rho_s u_{as}{}^2 r dr)}, \\
&= \frac{\int_0^{R_g} \rho_g u_{ag} u_{\theta g} r dr + \int_{R_g}^R \rho_s u_{as} u_{\theta s} r dr}{R(\int_0^{R_g} \rho_g u_{ag}{}^2 dr + \int_{R_g}^R \rho_s u_{as}{}^2 dr)}
\end{aligned} \tag{10}$$

where, $R$ is the radius of the gas column, $R_g$ is the density of the gas phase, $\rho_g$ is the axial velocity of the gas phase, and $u_{\theta g}$ is the tangential velocity of the gas phase. Similarly, $\rho_s$ is the density of solid particles, $u_{as}$ is the axial velocity of the solid phase, $u_{\theta s}$ is the tangential velocity of the solid phase, and r is the radial coordinate of the swirl pipe. The gas velocity set in this formula is the same as the velocity of the particle, and can be regarded as $u_{ag} = u_{as}$, $u_{\theta g} = u_{\theta s}$.

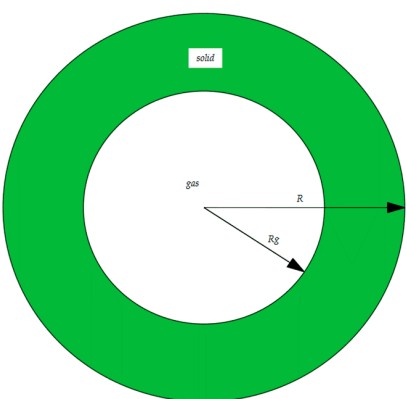

**Figure 6.** Swirl flow model diagram of gas-solid two-phase pipeline.

Sectional Solidity, $\alpha$:

$$\alpha = 1 - \frac{A_g}{A} = 1 - \frac{\pi R_g{}^2}{\pi R^2} = \frac{R^2 - R_g{}^2}{R^2}, \tag{11}$$

Substituting into the above formula (10), the finalization is simplified:

$$S = \frac{\rho_g u_a u_\theta R_g{}^2 + \rho_s u_a u_\theta (R^2 - R_g{}^2)}{2R\left[\rho_g u_a{}^2 R_g + \rho_s u_a (R - R_g)\right]} = \frac{u_\theta}{u_a} \frac{\left[\rho_g R_g{}^2 + \rho_s (R^2 - R_g{}^2)\right]}{\left[2\rho_g R_g{}^2 R + 2\rho_s R (R - R_g)\right]}$$
$$= \frac{(1 - \alpha)\rho_g + \alpha\rho_s}{2\sqrt{1 - \alpha}\rho_g + 2(1 - \sqrt{1 - \alpha})\rho_s} \frac{u_\theta}{u_a}, \tag{12}$$

The $\zeta$ may be written as:

$$\frac{(1 - \alpha)\rho_g + \alpha\rho_S}{2\sqrt{1 - \alpha}\rho_g + 2(1 - \sqrt{1 - \alpha}\rho_s)} = \zeta, \tag{13}$$

Then, the formula (12) becomes as follows:

$$S = \zeta \frac{u_\theta}{u_a}, \tag{14}$$

### 3.2. Cyclone Number Attenuation Law

Kreith Sonju, Wolfeta, and Baker Sayre [11] conducted an experimental study on the attenuation of swirl flow. Najafi [17] theoretically deduced the attenuation of swirl flow. The results show that the swirl attenuation law is expressed by the following formula:

$$S = S_0 \exp(-\beta \frac{L}{D}), \tag{15}$$

where, $S$ is the swirl number at different positions; $S_0$ is the swirl number at the initial position; $L$ is the distance from the initial position, (m); $D$ is the diameter of the pipe, (m); and $\beta$ is the attenuation index of the swirling intensity. The results of existing research show that in the horizontal straight tube swirl flow, the swirl number decreases exponentially along the axial direction.

### 3.2.1. Effect of Twist Rate on the Attenuation of Swirl Number

The variation law of the swirl number with different twist rates in different positions is shown in Figure 7. It can be seen from the figure that when $Z/D = 0$, the initial swirl number of the swirl flow with different twist ratios is different. When the twist rate is gradually increased, that is, the degree of rotation of the twisted belt is gradually weakened, the initial swirl number is gradually decreased. The twist band rate has an important influence on the initial swirl number of the swirl flow. In addition, the swirl number of the ordinary non-swirl (i.e., the twist band twist rate $Y = 0$) is always zero, and no change has occurred.

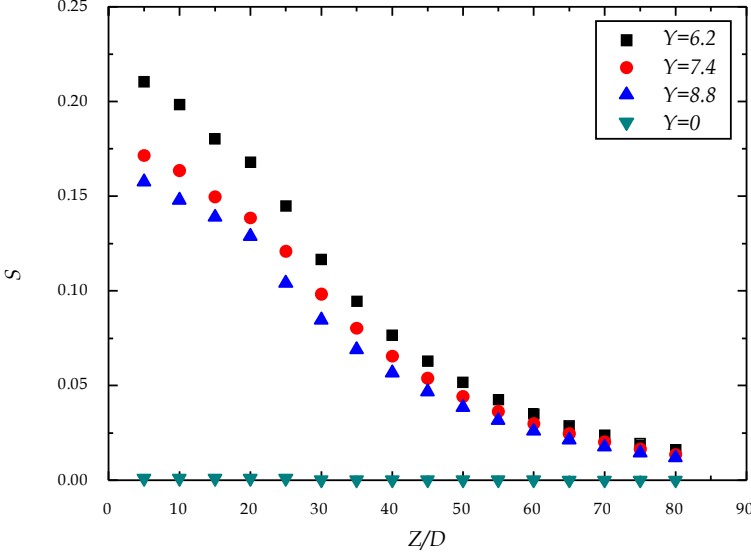

**Figure 7.** The swirl number at each position under different twist rates ($\alpha_0 = 2\%$, $Re = 20{,}000$).

### 3.2.2. Effect of Reynolds Number on the Decay of the Swirl Number

The variation law of the swirl number under different Reynolds numbers at different positions is given in Figure 8. As can be seen from the figure, when the twist band twist rate is the same, although the fluid Reynolds number is different, the initial swirl number, $S_0$, is also the same. Therefore, the Reynolds number has little effect on the initial Reynolds number of the swirl flow. It can be seen from the figure that as the swirl flow moves forward in the axial direction, the cross-section swirl number increases gradually as the Reynolds number increases.

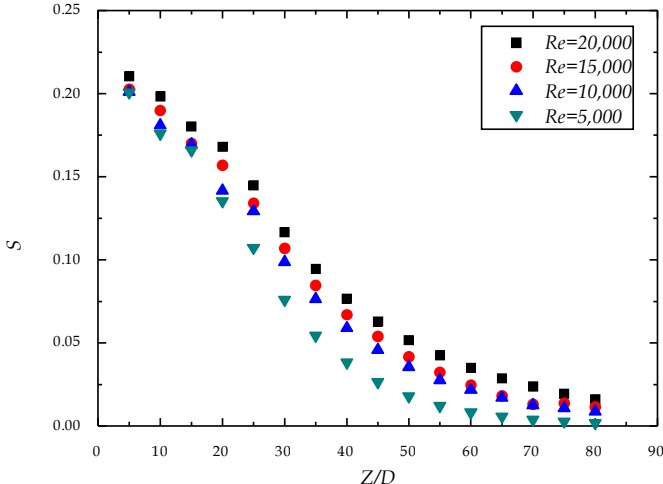

**Figure 8.** Variation in the swirl number at various positions under different Reynolds numbers ($\alpha_0 = 2\%$, $Y = 6.2$).

### 3.2.3. Effect of Hydrate Particle Concentration on the Decay of Swirl Number

The variation of the swirl number under different hydrate particle concentrations at different positions is shown in Figure 9. As can be seen from the figure, when the hydrate particle concentration is different, the swirl flow cross-section swirl number in the same position is the same. Therefore, the hydrate particle concentration has no effect on the initial swirl number of the swirl flow. In addition, as the swirl flow moves forward along the axis, the swirl number does not change with the increasing concentration of hydrate particles at the same location. Therefore, the concentration of hydrate particles has no influence on the convective swirl number. However, the attenuation law of the swirl flow number along the axial direction is consistent with the attenuation law of the swirl number under other conditions.

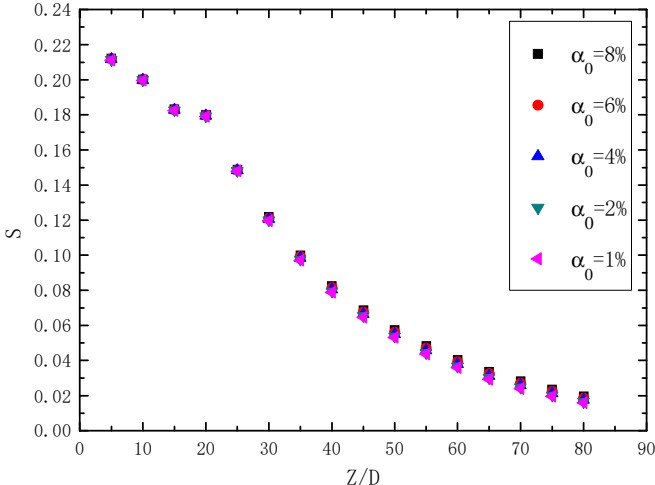

**Figure 9.** The swirl number at each position at different concentrations ($Y = 6.2$, $Re = 20,000$).

### 3.3. Cyclone Number Attenuation Law

The numerical simulation results, correlation curves, and literature analysis results in this paper are shown in Figure 10. It can be seen from the figure that the experimental results obtained by numerical simulation are basically the same as those of the literature [18]. Therefore, the predicted results basically meet the requirements.

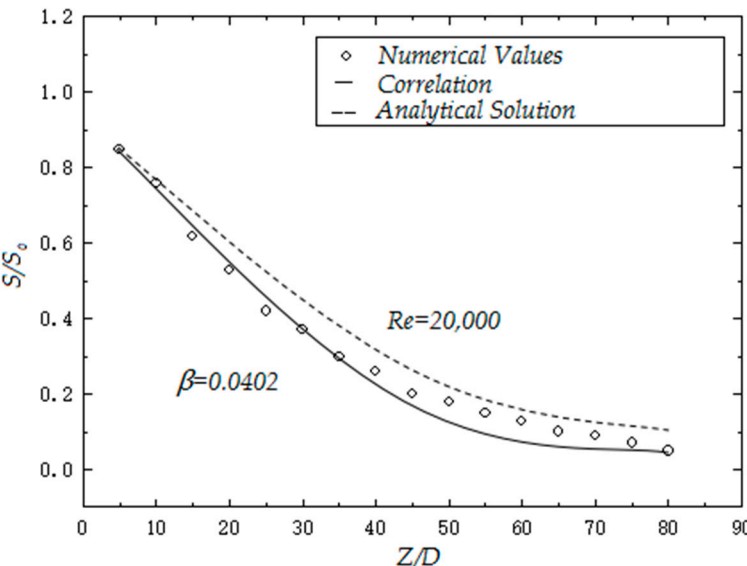

**Figure 10.** Comparison of numerical simulations with the results of the dimensionless swirl number in the literature.

### 3.3.1. Influence of Reynolds Number on the Decay Rate of the Swirl Number

There is a relationship between the swirl flow attenuation rate and the Reynolds number. To study the effect of Reynolds number on the attenuation rate of the swirl number, the author drew a graph of the relationship between the swirl flow decay rate, $\beta$, and the Reynolds number, $Re$, based on the simulated experimental data, as shown in Figure 11. The spiral flow decay rate $\beta$ and $ln(Re)$ are approximately inversely proportional.The spiral flow decay rate $\beta$ decreases with the increase of $ln(Re)$. The author thinks that when $ln(Re)$ was gradually increased, it means that the Reynolds number, $Re$, was gradually increasing, the swirl flow velocity was gradually increased, and the axial velocity and the tangential velocity were increased. Additionally, by keeping the tangential velocity at a relatively long-distance with a relative value, the long-distance maintenance of the swirl number was achieved, and the swirl flow decay rate, $\beta$, was reduced. Therefore, in order to maintain the swirl flow number at a higher level, it is necessary to appropriately increase the fluid Reynolds number. According to the results of the simulation experiment, the curve was obtained by least squares fitting, and the coefficient of $\beta = 0.339 - 0.03ln(Re)$ was obtained, which was the calculation method of the swirl flow decay rate, $\beta$, and the Reynolds number, $Re$.

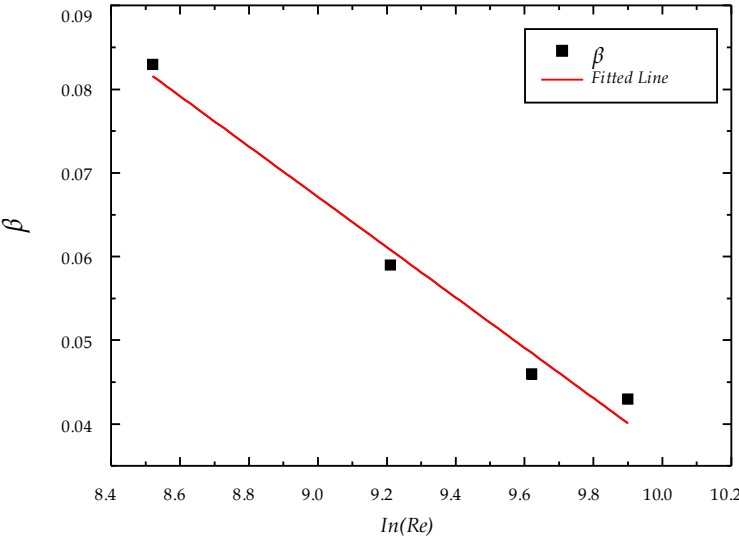

**Figure 11.** The relationship between the swirl flow decay rate, *β*, and the Reynolds number, *Re*.

3.3.2. Effect of Initial Concentration of Hydrate Particles on the Decay Rate of the Swirl Number

The initial concentration of hydrate particles will vary according to the operating conditions within the gas pipeline. To realize the safe operation of the gas pipeline, the author drew a graph of the relationship between the swirl flow decays rate, *β*, and the initial concentration, *α*, of the hydrate particles based on the simulated experimental data, as shown in Figure 12. The swirl flow decays rate, *β*, and the initial concentration, *α*, of the hydrate particles show a proportional relationship, although the swirl flow decay rate, *β*, gradually increases with the gradual increase of the initial concentration, *α*, of the hydrate particles. Thus, the effect of hydrate particle concentration on the swirl flow decay rate was minimal. The authors believe that when the initial concentration of hydrate particles gradually increased, the hydrate particles swirled forward in the pipeline due to the tangential velocity. The hydrate particles are light in weight and small in size, so they move forward in the axial direction without affecting the tangential speed [18]. When the concentration was gradually increased, the tangential velocity was related to the upper movement due to the need to carry more hydrate particles, thus causing a slight decrease in the tangential velocity, so the swirl number was gradually decreased, and the swirl flow decay rate, *β*, slightly increased.

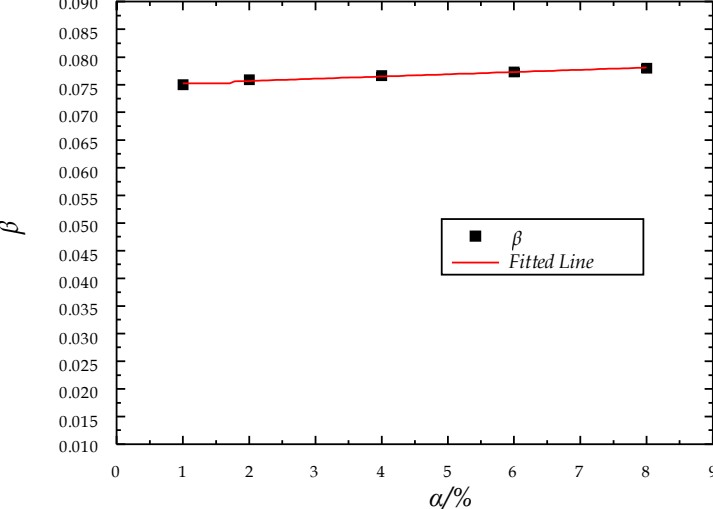

**Figure 12.** Relationship between the swirl flow decay rate, *β*, and the initial concentration, *α*.

### 3.3.3. Influence of Flow Distance on the Attenuation Rate of the Swirl Number

The effect of the flow distance on the swirl number is shown in Figures 13 and 14. Figure 13 shows the effect of the flow distance on the attenuation rate of the swirl number under three different twist rates. It can be seen from the figure that under the same Reynolds number, the attenuation rate of the swirl flow with different twist rates increases with the increase of the flow distance, and the decay rate of the swirl number decreases, and the decline law is basically the same. Therefore, the change of the twist rate has little effect on the attenuation rate of the swirl flow of the swirl flow, and the twist rate only affects the magnitude of the initial swirl flow. In general, as the flow distance increased, the attenuation rate gradually decreased, indicating that the farther away from the spinner, the smaller the attenuation rate of the swirl number, and the slower the attenuation of the swirl number. The author believes that the twist rate was different, only the initial swirl number was different, and the increase of the flow distance and the influence of the twist rate on the attenuation of the swirl flow in the subsequent pipeline was zero. Therefore, it is expressed that the swirling rate attenuation rate was the same. Figure 14 shows the effect of flow distance on the decay rate of the swirl number in the case of different Reynolds numbers. It can be seen from the figure that under the same twist ratio, the swirl flow rate under different Reynolds conditions increased with the increase of the flow distance, but the decline rate was different. As the flow distance increased, the attenuation rate also decreased gradually, indicating that the farther away from the spinner, the smaller the attenuation rate of the swirl number, and the slower the attenuation of the swirl number. Therefore, the change of the Reynolds number has a great influence on the attenuation rate of the swirl flow number, and the larger the Reynolds number, the smaller the coefficient, $\beta$, and the slower the swirl flow attenuation. The authors believe that the Reynolds number was different, the initial swirl number of the swirl flow was different, and the tangential velocity of the fluid along the path was also different. If the Reynolds number was large, the initial swirl number was large, and the swirl number along the path was also large, so the swirl number was slower and the decay rate was lower. The flow distance is also an important aspect that affects the attenuation rate of the swirl number. As the flow distance increased, the tangential flow velocity gradually decreased, the swirl number gradually decreased, and the attenuation rate gradually decreased. If the pipe was long enough and there were no relays in the process, the final decay rate was zero and the swirl flow eventually turned into a non-swirl.

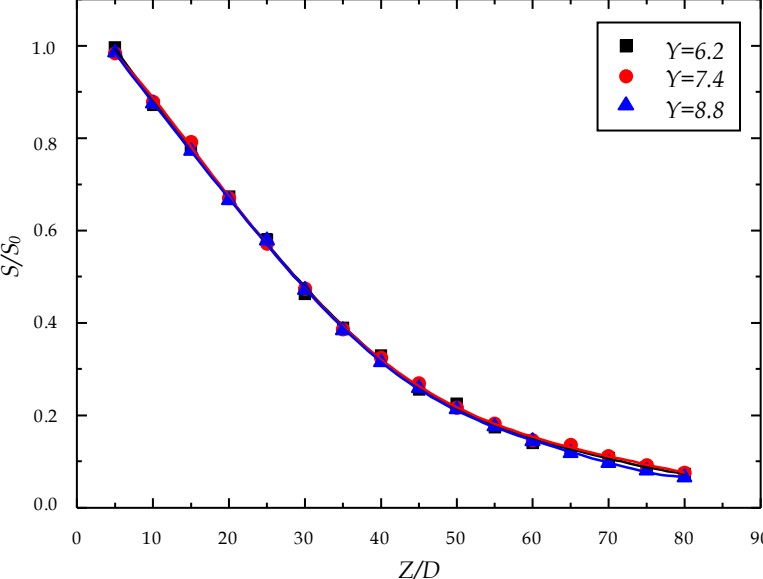

**Figure 13.** When *Re* = 20,000, the number of dimensionless swirl varies with the position at different twist rates.

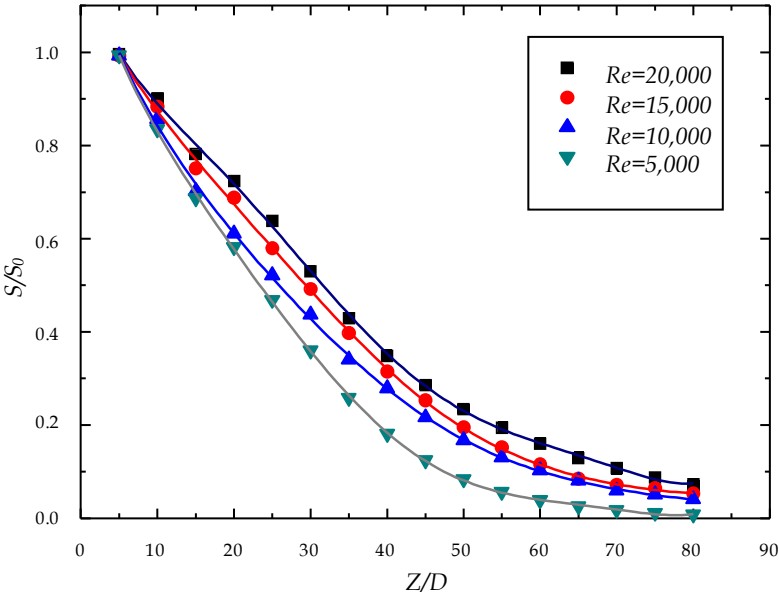

**Figure 14.** When $Y$ = 6.2, the number of dimensionless swirl varies with the position at different Reynolds numbers.

*3.4. Shear Force*

The forward movement of fluid in the pipeline is subject to various forces, including pressure, frictional resistance, shearing forces, etc., where the frictional shear forces generated by the fluid flowing through the pipeline have a greater impact on the hydrate particles. The existence of shear force has a significant influence on the variation of tangential velocity, the attenuation of the swirl number, and the deposition of hydrate particles. Therefore, studying the effects of the twist rate and Reynolds number on the shear force has a guiding role in elucidating the swirl motion trajectory of the hydrate particles and the deposition law.

3.4.1. Effect of Twist Rate on Shear Force

Under the condition of Reynolds number, $Re$ = 20,000, the influence law of the twist rate on the shear force is shown in Figure 15. It can be seen from the figure that as the flow distance increased, the frictional shear force gradually decreased. In the ordinary light pipe flow, the shear force change curve tended to the horizontal line and was close to zero. In the non-swirl state, the friction shear force of the fluid in the pipe was almost zero. In addition, as the twist rate decreased, the slope of the curve gradually increased, and the variation of the twisted band (before $Z$ = 40 $D$) was more obvious. The author believes that the characteristic tangential velocity of the swirl flow could cause the fluid to generate shear force. Due to the action of the twisted belt, a forced eddy current zone was generated in the pipeline, which enhanced the shear force of the hydrate particles and the wall surface of the pipeline. It could carry a large concentration of hydrate particles to move forward, and it was difficult for the hydrate particles to adhere to the inner wall of the pipe. In addition, hydrate particles were not easily deposited on the bottom of the pipe to achieve efficient and safe transportation of the hydrate particles. However, as the swirl flow gradually decayed, the shear force between the hydrate particles and the wall surface decreased, and the hydrate particles gradually deposited at the bottom of the pipe, causing blockage of the pipe in severe cases. Thus, to achieve efficient and safe transportation of natural gas hydrate, a swirl flow with a high swirl number in the pipe should be maintained.

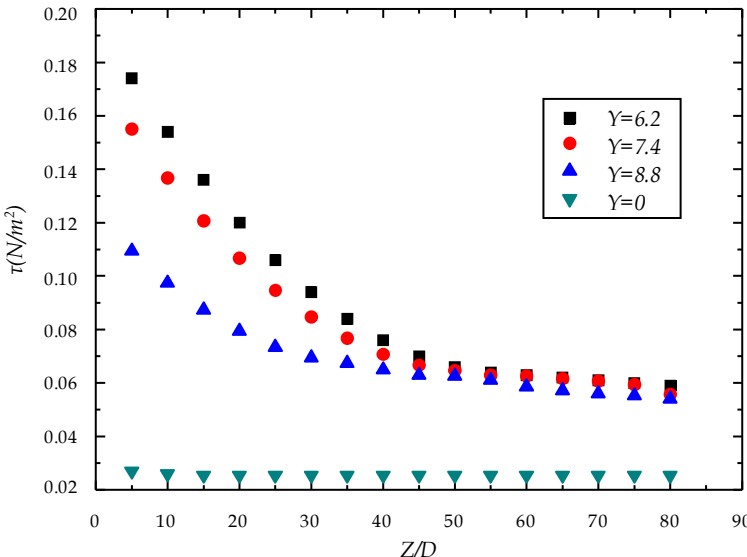

**Figure 15.** When *Re* = 20,000, the shear force changes at different positions of different twist rates.

### 3.4.2. Effect of Reynolds Number on Shear Force

When the twisting band twist rate was *Y* = 6.2, the influence law of Reynolds number on the frictional shear force is shown in Figure 16. It can be seen from the figure that as the flow distance increased, the frictional shear force also tended to decrease gradually; The initial shear force of the fluid in the pipeline was different when the Reynolds number was different; as the Reynolds number increased, the initial shear force of the fluid in the pipeline gradually increased. In addition, as the Reynolds number increased, the slope of the curve increased gradually, and the change of the twisted band (before *Z* = 40 *D*) was obvious, indicating that the larger the Reynolds number, the more the shear force decreases. The authors believe that, consistent with the above analysis, the tangential velocity characteristic of the swirl flow could cause shearing force in the fluid. Due to the action of the twisted band, a forced eddy current zone was generated in the pipe, which enhanced the shear force of the hydrate particle and the pipe wall. It could carry a large concentration of hydrate particles to move forward, so that it was difficult for the hydrate particles to adhere to the inner wall of the pipe, so that the hydrate particles were not easily deposited on the bottom of the pipe, and the hydrate particles were efficiently and safely transported. When the Reynolds number was gradually increased, the swirl flow had a large tangential velocity and a large swirl number. Therefore, the initial shear force of the pipe fluid was also relatively large, and it as easy to carry and transport the hydrate particles. The larger the Reynolds number, the greater the shear attenuation rate, that is, the swirl flow was more easily attenuated, the tangential velocity of the main flow of the swirl flow as also gradually reduced, and the hydrate particles were gradually deposited at the bottom of the pipe; this was not conducive to the safe transport of hydrate particles. In order to achieve safe and efficient delivery of hydrates, it is necessary to appropriately increase the Reynolds number of the fluid.

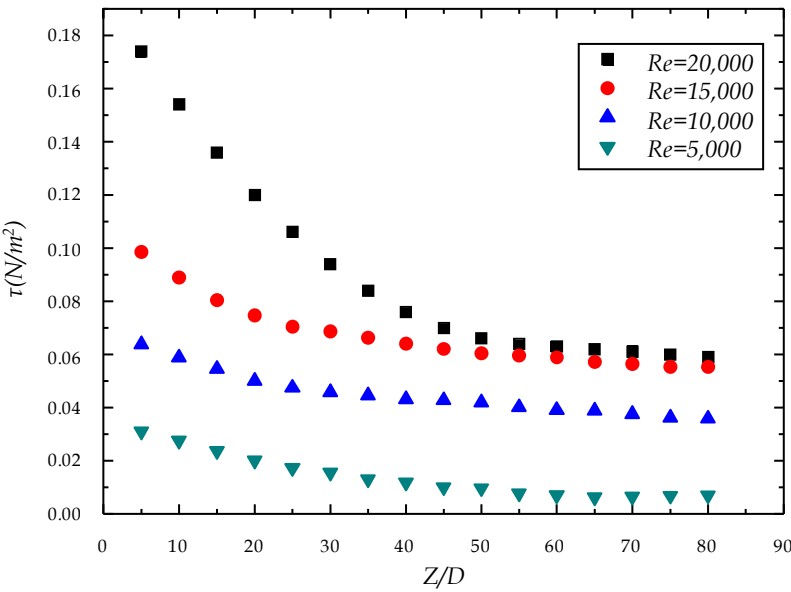

**Figure 16.** When *Y* = 6.2, the shear force changes at different positions of different Reynolds numbers.

### 3.5. Effect of Twist Rate on Volume Fraction Distribution of Hydrate Particles

Under the condition of Reynolds number, *Re* = 20,000, the effect of the twist band rate on the volume fraction of hydrate particles is shown in Figure 17. It can be seen from the figure that at the same position, when the twist rate was zero, there was no swirl flow in the pipe, and only the non-swirl flow moved forward, the hydrate volume fraction in the upper part of the pipe was almost zero, and the hydrate particle concentration in the lower part of the pipe was a substantial increase. It indicates that when there was no swirl flow in the pipeline, the pipeline hydrate particles gradually deposited to the bottom of the pipeline. With the gradual increase of the flow distance, under the condition of swirl flow, hydrate particles were gradually deposited in the upper part of the pipe, and the degree of hydrate particle deposition in the lower part of the pipe was gradually increased. In general, along with the direction of the pipe, the particle volume fraction at the bottom of the pipe became larger and the deposit became more [19]. The authors believe that the light pipe had almost no particles in the upper half of the pipe section compared to the twisted pipe because the particles were deposited by gravity at the bottom of the pipe. The tangential velocity generated by the twisting of the twisted belt had a strong carrying capacity, and the hydrate was gradually distributed around the pipeline due to the carrying of the swirl flow. The hydrate particles in the center of the pipe were gradually reduced, and the closer to the pipe wall, the larger the volume fraction of the particles. In addition, as the twisting rate of the twisted band was reduced, the degree of rotation of the twisting rate was increased, and the carrying capacity of the hydrate particles was greatly improved. Therefore, the smaller the twisting rate, the less the particle deposition at the bottom of the tube. In general, the swirl flow generated by the twist band was placed in the pipeline, so that the hydrate particles were carried a larger distance and were not easily deposited on the pipe wall, which was beneficial to the safe transportation of the hydrate particles.

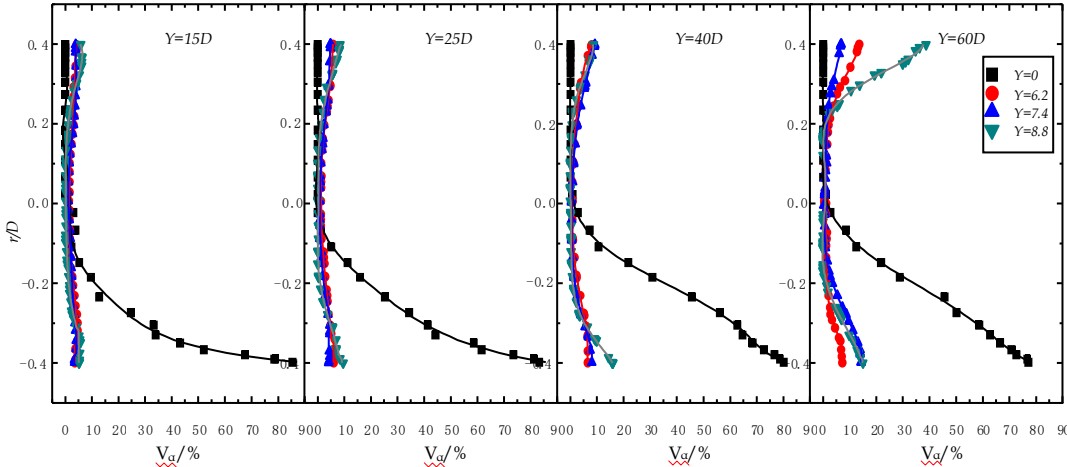

**Figure 17.** Graph of particle volume fraction at each section under different twist conditions at *Re* = 20,000.

## 3.6. Effect of Particle Volume Concentration on Hydrate Particle Deposition

The hydrate particle deposition rule of different particle concentrations under the Reynolds number, *Re* = 20,000, and the twist rate of 6.2 is shown in Figure 18. At the same position, as the concentration of hydrate particles increased, the particle concentration at the top and bottom of the pipe also increased. Therefore, the increase in the concentration of hydrate particles under the swirl flow system caused the accumulation of particles at the top of the pipe and the deposition of particles at the bottom of the pipe. With the gradual increase of the flow distance, the degree of deposition of hydrate particles at the top and bottom of the pipe gradually increased under the condition of swirl flow, and, therefore, the deposition of hydrate particles was gradually increased due to the attenuation of the swirl flow. In general, an increase in particle concentration increased the risk of deposits and blockage of hydrate particles in the gas pipeline [20].

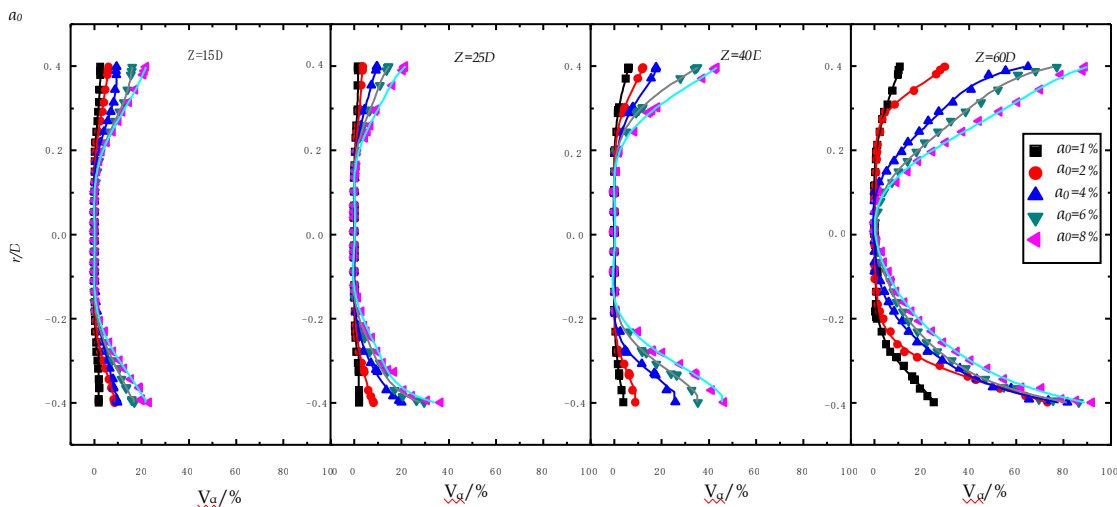

**Figure 18.** Graph of particle volume fraction at each section at *Re* = 20,000 with a twist ratio of 6.2.

The authors believe that the hydrate particle concentration was an important factor affecting the transport and deposition of hydrate particles. According to the simulation results, to achieve the efficient transport of hydrate particles, the initial concentration of hydrate particles in the gas pipeline should be strictly controlled. It has guided significance for controlling hydrate blockage in pipelines and can provide a guarantee for safe transport of hydrates.

## 4. Conclusions

In this paper, the numerical simulation method was used to study the flow and attenuation of hydrate particles under the swirl flow system in gas pipelines. The simulation results were compared with the experimental data to provide guidance for the study of hydrate safe flow.

The initial swirl number was only related to the twist rate and was independent of the Reynolds number and concentration. As the twist rate increased, the swirl number decreased, and the swirl number and the twist rate showed an exponential relationship. The swirl number along the pipe increased as the Reynolds number increased, and the concentration had little effect on the swirl number.

The swirl flow attenuation coefficient, $\beta$, and $ln(Re)$ were close to a proportional relationship. The main effect of the swirl flow attenuation was the Reynolds number, Re, and the larger the Reynolds number, the smaller the coefficient, $\beta$, and the slower the swirl flow attenuation. The relationship between the concentration and the decay rate, $\beta$, was close to a horizontal line, and the concentration had little effect on the swirl flow attenuation.

In ordinary light pipe flow, the shear force change curve tended to be horizontal and close to zero, and hydrate particles were easily deposited in the pipe. As the twist rate decreased, the slope of the curve gradually increased and the shear force also became larger. The change of the Reynolds number had a great influence on the shear force. The larger the Reynolds number, the larger the shear force and the less likely the hydrate particles were deposited.

The twist-free belt was six to eight times larger than the volume fraction of the twisted belt, and the smaller the twisted twist ratio, the less the bottom particle deposition. The twisted band was placed in the pipeline, and the particles carried a larger distance and were not easily deposited on the pipe wall, which is beneficial to the safe transportation of the hydrate particles. If the spinner is improved, the turbulent energy is not easily attenuated, the hydrate particle blending effect is better, and the transport distance is further. If the initial particles concentration was higher, the volume fraction of hydrate particles deposited was larger. Controlling the initial concentration of particles has a guiding significance for controlling the clogging problem in the pipeline, and provides a guarantee for the safe transport of the hydrate.

**Author Contributions:** Conceptualization, R.Y. and W.S.; Methodology, R.Y. and W.S.; Validation, R.Y. and W.S.; Formal Analysis, R.Y., S.Y. and W.S.; Investigation, R.Y., S.Y. and J.R.; Resources, W.S.; Data Curation, W.S.; Writing-Original Draft Preparation, R.Y., S.Y. and J.R.; Writing-Review & Editing, R.Y., S.Y. and J.R.; Visualization, R.Y. and S.Y.; Supervision, W.S.; Project Administration, W.S.; Funding Acquisition, W.S.

**Funding:** The financial support from National Nature Science Foundation of China under the contract No. 51574045 is appreciated.

**Acknowledgments:** Thanks to Shuli Wang and Minguan Yang for having provided the workstation and technical guidances.

**Conflicts of Interest:** The authors declare no conflict of interest.

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
