# Peer review of "Numerical Simulation Study on the Law of Attenuation of Hydrate Particles in a Gas Transmission Pipeline"

_energies, doi:10.3390/en12010058_

Round 1

Reviewer 1 Report

The authors numerically studied the attenuation of swirl flow of hydrate particles in gas pipelines. They investigated the effect of twist rate, Reynolds number and concentration of hydrate particles on the decay of swirl characteristics of the flow. 

The authors did a competent job of discussing their results with proper justifications. I do not have any major concern with the discussed results. 

However, I would recommend the authors to kindly edit the paper sincerely to rectify multiple typos and use concise sentences to effectively communicate the explanations/ justifications to the readers. 

Author Response

Response to Reviewer 1 Comments

Point 1: The authors numerically studied the attenuation of swirl flow of hydrate particles in gas pipelines. They investigated the effect of twist rate, Reynolds number and concentration of hydrate particles on the decay of swirl characteristics of the flow.

Response 1: Dear reviewers and editors, thank you very much for your valuable suggestions and comments. Thank you very much for your agreement of the research.

Point 2: The authors did a competent job of discussing their results with proper justifications. I do not have any major concern with the discussed results.

Response 2: Dear reviewers and editors, thank you very much for your valuable suggestions and comments. Thank you very much for your agreement of the research again.

Point 3: However, I would recommend the authors to kindly edit the paper sincerely to rectify multiple typos and use concise sentences to effectively communicate the explanations/ justifications to the readers.

Response 3: Dear reviewers and editors, thank you very much for your valuable suggestions and comments. I have rechecked the manuscript, and found a lot of the typos and grammatical errors. I have checked grammar, spelling, punctuation and some improvement of style. I think the current language writing may meet the writing requirements of your Journal.

Reviewer 2 Report

major comments:

was the material of which the pipeline is composed somehow included in the calculations? In terms of possible adsorption?

It is not clear how the D and L values have been chosen?

minor comments:

It is very hard to write a review when the lines are not numbered,

Tha language is below the quality of the good scientific journal, some examples below:

Page 1: gas not Gas

Page 7: experimental experiments?

there is a lack of space between words in many places in the manuscript

Figure 10: should it refer to [16] or [17]?

Figure 12: the Y scale should be rescaled

Page 15: the experiments are compared with experiments? This make no sense...

Author Response

Response to Reviewer 2 Comments

Point 1: was the material of which the pipeline is composed somehow included in the calculations? In terms of possible adsorption?

Response 1: Dear reviewers and editors, thank you very much for your valuable suggestions and opinions. First of all, this paper focuses on the flow characteristics of natural gas hydrate particles and the attenuation law of swirl flow in the pipeline. Therefore, the main content of the research is the flow characteristics in the pipeline. In addition, there is no adsorption between the hydrate particles and the inner wall of the pipe (the pipe material is mostly X70 steel) in the actual gas pipeline. The effects of wall adsorption on hydrate flow characteristics were not considered in experiments and simulations. Please check and approve, thank you!

Point 2: It is not clear how the D and L values have been chosen?

Response 2: Dear reviewers and editors, thank you very much for your valuable suggestions and opinions. The authors built an experimental platform for simulating the flow of hydrate particles in the laboratory. The inner diameter of the pipe was 0.024m (D) and the length of the pipe was 2.5m (L). For pipe diameters, the author selected it according to the diameter of the most commonly used pipes in the industry. For the pipe length, the hydrate flow swirl attenuation distance is estimated in advance. The experimental data obtained from the experimental platform are used to compare and verify the numerical simulation results, which guides the simulation better. Please check and approve, thank you!

Point 3: It is very hard to write a review when the lines are not numbered.

Response 3: Dear reviewers and editors, thank you very much for your valuable suggestions and opinions. I am sorry that the problem has been caused by the lack of numbering for your review. I have already numbered the article. Please check and approve, thank you!

Point 4:The language is below the quality of the good scientific journal, some examples below:

Page 1: gas not Gas

Page 7: experimental experiments?

there is a lack of space between words in many places in the manuscript

Figure 10: should it refer to [16] or [17]?

Figure 12: the Y scale should be rescaled

Page 15: the experiments are compared with experiments? This make no sense...

Response 4: Dear reviewers and editors, thank you very much for your valuable suggestions and opinions. I have revised the paper according to your suggestion.. In addition, I have rechecked the whole manuscript, and found a lot of the typos and grammatical errors. I have checked grammar, spelling, punctuation and some improvement of style. I think the current language writing may meet the writing requirements of your Journal. Please check and approve, thank you!

Round 2

Reviewer 2 Report

none